# Condensed and Hydrolyzable Tannins for Reducing Methane and Nitrous Oxide Emissions in Dairy Manure—A Laboratory Incubation Study [note 1]

**DOI:** 10.3390/ani12202876

**Published:** 2022-10-21

**Authors:** Byeng Ryel Min, Will Willis, Kenneth Casey, Lana Castleberry, Heidi Waldrip, David Parker

**Affiliations:** 1Conservation and Production Research Laboratory, United States Department of Agriculture (USDA)/Agricultural Research Service (ARS), Bushland, TX 79012, USA; 2Department of Agricultural and Environmental Sciences, Tuskegee University, Tuskegee, AL 36088, USA; 3Texas A & M AgriLife Extension Center, Amarillo, TX 79106, USA

**Keywords:** dairy cattle manure, greenhouse gas, methanogens, microbiome

## Abstract

**Simple Summary:**

This study investigated the mitigation effect of different tannins on greenhouse gas emissions (GHG) from freshly collected dairy cattle manure. Quebracho and chestnut industrial tannin extracts were tested against a fresh cattle manure slurry. The results showed that all tannins added to freshly collected dairy manure could decrease N_2_O and CH_4_ emissions over a 14-d incubation period.

**Abstract:**

The objectives of this study were to (1) examine the effects of plant condensed (CT) and hydrolyzable tannin (HT) extracts on CH_4_ and N_2_O emissions; (2) identify the reactions responsible for manure-derived GHG emissions, and (3) examine accompanying microbial community changes in fresh dairy manure. Five treatments were applied in triplicate to the freshly collected dairy manure, including 4% CT, 8% CT, 4% HT, 8% HT (V/V), and control (no tannin addition). Fresh dairy manure was placed into 710 mL glass incubation chambers. In vitro composted dairy manure samples were collected at 0, 24, 48, and 336 h after the start of incubation. Fluxes of N_2_O and CH_4_ were measured for 5-min/h for 14 d at a constant ambient incubation temperature of 39 °C. The addition of quebracho CT significantly decreased the CH_4_ flux rates compared to the tannin-free controls (215.9 mg/m^2^/h), with peaks of 75.6 and 89.6 mg/m^2^/h for 4 and 8% CT inclusion rates, respectively. Furthermore, CT significantly reduced cumulative CH_4_ emission by 68.2 and 57.3% at 4 and 8% CT addition, respectively. The HT treatments failed to affect CH_4_ reduction. However, both CT and HT reduced (*p* < 0.001) cumulative and flux rates of N_2_O emissions. The decrease in CH_4_ flux with CT was associated with a reduction in the abundance of Bacteroidetes and Proteobacteria.

## 1. Introduction

The livestock industries produce greenhouse gases (GHG) in the form of methane (CH_4_) from enteric fermentation, nitrous oxide (N_2_O) from the use of nitrogenous substrates, and CH_4_ and N_2_O from manure management and deposition of animal manures on pastures [1]. Animal agriculture contributes an estimated 8 to 18% of the total global anthropogenic GHG emissions [1,2,3], including CH_4_ and N_2_O. For instance, the global warming potential (GWP) of CH_4_ and N_2_O have been computed as 28- and 265-times, respectively, that of CO_2_ for a 100-year timescale [4]. Agriculture contributes about 10% of total U.S. GHG emissions, but livestock contributes about 4% of total U.S. GHG emissions, excluding feed production and fuel use [5]. The livestock-related GHG emissions will increase as the world population and food demand increase [1]. 

Tannins have occasionally been regarded as an “anti-nutritional factor” for poultry and other non-ruminants. More recent research revealed that when used appropriately in cattle diets, some tannins improved gut microbial ecosystems, enhanced animal performance, and reduced enteric and manure-derived GHG emissions [6,7,8]. Tannins can be classified into two groups: (1) hydrolyzable tannins (HT), which are primarily comprised of gallic acid building blocks linked to sugar by esterification; and (2) condensed tannins (CT), which are comprised of flavan-2-ols monomer (or proanthocyanidins) linkages [9]. Plant tannins interact with proteins and other molecules, thereby playing a role in soil and grassland ecosystems through their influence on plant biomass [10], allelopathy [11], and carbon (C) and nitrogen (N) cycling [12]. In addition, plant tannin-containing diets or extracts have been proposed to reduce enteric CH_4_ emissions from ruminants [13,14] and ammonia (NH_3_) emissions from fresh manure [15,16,17]. Plant tannins inhibited rumen and fecal microbial activities [18,19], formed complexes with dietary protein and carbohydrates [6,20], and reduced ruminal CH_4_ production [16]. In addition, direct application of quebracho CT extracts to compost (30% Rhodes grass hay (*Chloris gayana* K.) + 70% farmyard manure) reduced cumulative gas emissions by 40% and N_2_O emission by 36%, compared with the non-amended manure compost [21]. Published results suggest that CT has more inhibitory activity against Gram-positive than Gram-negative fecal bacteria [22,23] because Gram-positive bacteria predominate the cattle manure microbiome, including genera of *Clostridium* (14.4 %), *Bacteroides* (11.3%), and the order *Bacteroidales* (8.4%; [24]). Therefore, the goal of the current research was to investigate the effects of the addition of two sources of tannins (quebracho (CT; *Schinopsis* sp.) versus chestnut (HT; *Castanea sativa*)) at three different dose levels on GHG emissions and microbial activities in an in vitro real-time culture system of incubated dairy cattle manure. The effects were determined by evaluating cumulative CH_4_ and N_2_O gas production and 16S rDNA analysis for microorganism diversity changes.

## 2. Materials and Methods

### 2.1. Experimental Design

Cows generate CH_4_ in two main ways: their digestion and waste. This study only investigated cow waste (fresh manure) to investigate manure-derived GHG emissions. An in vitro experiment was conducted to determine the effect of two tannins and three different dosage levels on the chemical properties of the dairy cattle manure and its GHG emissions, both of which are associated with microbiome changes from freshly collected dairy cattle manure. In addition, a sub-set of incubation chambers was prepared to collect data for moisture, chemical, and microbiological sampling throughout the experiment. The separate chambers were necessary to avoid disrupting the emitting surface in chambers connected to the real-time monitor. Airflow rate, temperature, and other conditions for the sampling changes were identical to those of the emission monitoring chambers. The study was conducted from 8 to 22 April 2019 using in vitro incubation at the USDA/ARS, Bushland, TX. Commercially available quebracho and chestnut tannins were purchased (Chemtan Co. Inc., Exeter, NH, USA) and used as sources of CT and HT, respectively. The experimental design was a 2 × 3 multiplication design with two different sources of tannins (CT vs. HT) and three different levels of tannins. There were three replicates per treatment.

### 2.2. Fresh Dairy Manure Substrate Preparation

Twenty fresh dairy cattle manure was obtained by randomly collecting fresh feces (200 g/cattle) from 2000 commercial dairy cattle in the Texas Panhandle. The fresh feces were mixed well in the laboratory to make the stock manure for each treatment combined as a batch and then distributed to each of the three replicates. Five treatments were applied in triplicate (15 chambers total) to the manure, including 4% CT, 8% CT, 4% HT, 8% HT (v/v), and control (no tannin addition). Chamber temperatures were controlled with vented thermal blankets. Manure (220 g) was placed into 710 mL glass incubation chambers. The manure sample was 74.6 mm deep by 76.2 mm in diameter. Two empty chambers served as system blanks, and consistent background levels of CH_4_ and N_2_O changes in the blanks were not detected in the absence of manure. Chambers were maintained at 39 °C for the duration of the experiment. Sub-samples of manure before (0 h) and after (336 h) incubations from the same chambers in each treatment were also collected (2 × 20 g) for chemical and microbiome analyses.

### 2.3. Gas Production Analysis

Each container was connected to a multiplexer and a Los Gatos real-time N_2_O analyzer. Each container was connected through two pieces of Excelon Bev-A-Line IV^®^ Tubing (United States Plastic Corp, Lima, OH, USA) inside diameter 3.1 mm (0.125 in). When a single container was sampled, the sampling air was recirculated at a rate of 400 mL/min between the container and the analyzer. With this closed system, the concentration of N_2_O in the sampling loop increased over time. Clean ambient air was passed through the other 15 containers at a rate of 40 mL/min when not being sampled. A container was sampled every 3 min 45 s, so every container including blank chambers was sampled once per hour. Following a 165-s flushing period, the next chamber was sampled. During the 60-s sampling period, the multiplexer valves were adjusted in a way so the container became a small recirculating flow through—a non-steady-state (RFT-NSS) chamber (i.e., a static chamber system) with CH_4_ and N_2_O concentrations being measured every 2 s. 

Chamber temperature and days of incubation were chosen at 39 °C for the present study because Waldrip et al. [25] indicated that most N_2_O emissions occurred (>95%) after a simulated rainfall event (25 mm) within 7-d at temperatures ranging from 27 °C to 46 °C. A programmable multi-valve multiplexer was used to switch measurements among chambers. When in non-measurement mode, ambient air was continuously passed through all flux chambers to prevent the accumulation of GHG or the creation of artificial anaerobic conditions. When in flux measurement mode, the flux chambers were operated as static, non-steady-state flux chambers. The multiplexer was controlled by a Campbell Scientific CR6 datalogger/digital controller using Loggernet software (Campbell Scientific, Logan, UT, USA). Nitrous oxide and CH_4_ concentrations were monitored using real-time analyzers (Los Gatos Fast GHG Enhanced Performance Analyzer for CH_4_/CO_2_/H_2_O; 3055 Orchard Dr., San Jose, CA, USA).

### 2.4. Sampling and DNA Extraction from In Vitro Dairy Cattle Manure

After the 14-d incubation period was completed, chambers and remaining manure were weighed to measure moisture levels. The subsamples (20 g each) were collected from the top 2 cm (0–2 cm) and the bottom portion (2–4 cm) and combined to form one sample, immediately covered, and stored at −80 °C until microbiome analysis. All manure samples were thawed at 4 °C, and transferred on ice, and mixed with an equal volume of the substrate with 10% (v/v) zirconium beads (0.1 mm). Samples were homogenized for 30 s at a speed of 4 m/s in a FastPrep Instrument (Q-BIOgene, Irvine, CA, USA), followed by centrifugation for 10 min at 14,000× *g*. A 600-µL aliquot of the supernatant was then added to 500-µL Fastprep Binding Matrix and extracted using a FastDNA Spin Kit (Q-BIOgene; MP Biomedicals, Santa Ana, CA, USA) according to the manufacturer’s instructions. DNA extractions were performed on approximately 100 mg of the collected pellets using a Quick-DNA Fungal/Bacterial Miniprep Kit (ZymoResearch, Irvine, CA, USA) according to the manufacturer’s protocol. DNA was quantified using a Nanodrop spectrophotometer (Nyxor Biotech, Paris, France). This purified DNA sample was investigated for bacterial diversity using a prokaryotic tag-encoded FLX amplicon pyrosequencing (FLX bTEFAP) PCR method [26]. Data quality control and analyses were directed as described by Dowd et al. [26]. A HotStarTaq Plus Master Mix Kit (Qiagen, Valencia, CA, USA) was used for PCR over the following conditions: 94 °C for 3 min followed by 32 cycles of 94 °C for 30 s; 60 °C for 40 s, and 72 °C for 1 min; and a final elongation step at 72 °C for 5 min.

### 2.5. Manure Chemical Analysis

Manure samples were collected immediately before and after the completed experiments. Each sample was analyzed for water content, nitrate + nitrite-N (NO_3_^−^-N + NO_2_^−^-N), NH_4_^+^-N, total N, and pH by Servi-Tech Laboratories (Amarillo, TX). Total-N and organic-N were determined using the Kjeldahl method [27]. Ammonium was determined by titration in accordance with Standard Method 4500 [28]. Nitrate and -NO_2_ were analyzed by colorimetric flow injection analysis (FIA) according to EPA Method 353.2 [29]. Water content was determined using Standard Method 2540 by loss on drying at 105 °C for 24 h, and volatile solids were determined by loss on ignition at 500 °C [30]. Manure pH was determined on a 1:1 ratio of water to manure using probe meters according to the standard methods of 9050 and 9045D, respectively [31,32].

### 2.6. Statistical Analyses

The concentration vs. time curve slope for each monitoring period was calculated using linear regression for the 2-min period starting 30 s after closing the lid. Sixty data points were used in the regression [33,34]. Methane flux (mg CH_4_/m^2^/h) and N_2_O flux (mg N_2_O/m^2^/h) were calculated as the product of the slope of non-steady-state increase in concentration with time (mg/m^3^/min or h), and the effective headspace height (m). Headspace depth was 7.7 cm in a 710 mL jar (headspace was calculated from jar diameter (75 mm) and headspace height (92.075 mm)). Statistical analyses of GHG (CH_4_ and N_2_O) flux data included calculations of slopes, correlation coefficients, and regression coefficients using Excel (Microsoft Corp., Redmond, WA, USA) and the SAS PROC REG procedures [35]. Cumulative CH_4_ and N_2_O emissions were calculated in Excel by numerically integrating the area under the flux vs. time curves. The relative abundance of microbial community diversity and incremental gas production analyses were conducted using the GLM procedure of SAS Institute (SAS Inst. Inc., Cary, NC, USA). Factors examined included source of tannins (CT vs. HT), dose levels (0, 4, and 8% CT and HT), and tannin x dose levels interactions. Data were presented as least-squares mean, together with the standard error of the mean (SEM). The least-squares means are reported throughout, and significance was declared at *p* < 0.05.

## 3. Results and Discussions

### 3.1. Ambient Temperature, Chemical Composition of Dairy Cattle Manure

Initial and final moisture contents are presented in Table 1. The initial and final manure moisture content ranged from 59.8 to 62.7% and 23.6 to 31.8%, respectively. The chemical composition of control (no-tannins), quebracho CT, and chestnut HT before (d 0) and after 14-d in vitro incubation experiments are presented. Moisture, total solids, ash, organic matter, total N, NH_4_^+^-N, NO_3_-N, and pH on d 0 were similar among treatment groups, but the C/N ratio was generally greater for CT and HT groups than for the control group. Only trace amounts of NH_4_^+^-N were detected in dairy cattle manure. After the manure was incubated, with or without tannins, at 14 d, total N (%), organic-N (%), NH_4_-N, and NO_3_^−^/NO_2_^−^ (mg/kg) were similar among treatments. 

There is evidence that plant tannins can impact the long-term restriction of soil-N through the formation of tannin-protein complexes and the inhibition of enzyme activity in soils [36,37]. The addition of CT in the diets considerably shifted partitioning in N from urine to feces, which is in line with the results of the studies on CT effects in ruminant animals [38,39] and could contribute to lower ammonia emissions from manure in ruminant production [17]. It has been reported that supplementation of tannin extracts from quebracho and/or chestnut reduced NH_3_-N emissions by 30.6 to 51% by decreasing urease activity when these tannins were mixed with manure [17,21]. In the present study, the NH_4_-N content of the manure at the end of the 14-d incubation increased by 91.7 to 92.9% of the values at d 0, similar to Hao and Benke [40]. However, NH_4_-N represents a tiny proportion of the initial and final total N in the manure. In addition, total N loss was 80% of initial total N and appeared to be similar among the treatments. However, the exact nature of tannin’s inhibitory effects on urease activity and subsequent reductions in CH_4_ and N_2_O emissions remain unclear. 

### 3.2. Microbial Communities of Dairy Manure

Results of the manure bacterial phylum, species, and genera are presented in Table 2 and Table 3. Representative sequences from the operational taxonomic units (OTUs) were assigned to seven major bacterial phyla (relative abundance > 1.0%) as dominant, regardless of the treatment group. Some studies reported that Firmicutes and Bacteroidetes were the predominant phyla in the microbiota of all domesticated ruminants [18,19,24]; however, this trend has not been universal for other studies [41,42,43]. In the present study, the most abundant phyla detected in the control groups were Proteobacteria (32.8%), followed by Bacteroidetes (28.8 %), Firmicutes (14.1%), Actinobacteria (13.7%), Planctomycetes (7.0%), Chloroflexi (1.3%), and Euryarchaeota (1.1%) in dairy manure samples (Table 2). There was a change in the predominant bacterial populations, confirming that tannin affects the manure microbiome. The results showed a decrease (*p* < 0.04) in the proportion of Bacteroidetes in samples from dairy cattle manure incubated when the CT or HT was at the highest inclusion levels compared to control incubations. There tended to be a decrease (*p* = 0.09) in the proportion of Proteobacteria in samples incubated with HT compared to the control group. These results were similar to trends found in the rumen of sheep and goats [44] and the feces of tannin-fed rats [22]. In contrast, the level of Actinobacteria increased (*p* < 0.01) in the presence of both CT and HT compared to the control group. Similar tannin–dependent increases in select phyla were previously reported as tannin-resistant bacterial populations increased in humans [45], pigs [46], rats [22], chickens [47], and ruminants [48,49,50]. All these studies confirmed that tannins cause a shift in the manure microbiome, and a consistent result was that Gram-positive Firmicutes and Actinobacterial-type (actinomycetes) bacteria were enhanced. Determining the mechanisms by which bacteria can resist the inhibitory effects of tannins is essential to successfully implement a strategy of increasing the proportion of tannin-resistant bacteria in manure.

At the species-level analysis, using a 16S rDNA sequencing detected 588 classifiable bacterial species in all samples (Table 3), with 15 being dominant species (>1.0%). The source of tannins × dosages interactions was significant for *Proteiniphilum* spp. (*p* < 0.03) and *Nonomurae* sp. (*p* < 0.01) species, suggesting that higher CT addition rates decreased select bacterial concentrations when CT was mixed with dairy cattle manure but did not differ between 4% HT and the control. Across the dose level of tannins, tannins in addition to dairy cattle manure decreased the relative abundance of *Clostridium* spp. (*p* < 0.01), *Pusillimonas noertemannii* (*p* < 0.01), *Idiomarina indica* (*p* < 0.001), and *Mycoplana* spp. (*p* < 0.05) species, but *Clostridium* spp. were increased at 8% HT and reduced at only 4% HT. In contrast, populations of *Shigella sonnei* (*p* < 0.03), *Bifidobacterium choerinum* (*p* < 0.05), and *Corynebacterium* sp. (*p* < 0.05) were increased in tannin supplementation (Table 3), indicating that these microbial populations may be dependent upon the increased tannin levels in the cattle manure compared to the control manure. Plant tannins are secondary metabolites that function as part of a plant’s biological defense mechanism against invasion by pathogens and attack by insects. Moreover, plant tannins also interact with manure and soil nutrients and affect the microbial diversity changes and chemical processes that may be important for generating GHG emissions and nutrient cycling [8,46,51]. Likewise, the antimicrobial activities of tannins have long been recognized, and the toxicity of tannins to bacteria, fungi, and yeasts has been reviewed [52,53]. In vitro and in vivo studies have consistently shown a reduction in the growth rate of select strains as a consequence of dietary CT [7,54,55]. However, some strains (*Clostridium proteoclasticum* B316^T^ and *R. albus* 8) showed transient increases in their growth rate at low concentrations (50–100 µg/mL) but not at high (>200 µg/mL) concentrations of CT [55]. Generally, tannins act by inhibiting the microbial activity of Gram-positive bacteria; however, some studies indicate that tannins may be more effective against Gram-negative bacteria [22,56]. Nevertheless, it has been reported that pathogenic Gram-negative bacteria, such as *Escherichia coli* O157:H7, *Salmonella, Shigella*, *Pseudomonas*, and *Helicobacter pylori,* were sensitive to CT [43,57]. The number of hydroxyl groups (OH) and release of hydrogen peroxide (H_2_O_2_) upon oxidation of plant tannins are critical elements for the antimicrobial properties of tannins [22,58]. Interestingly, *B. choerinum* abundance increased with higher rates of both tannins (Table 3). It has been reported that *B. choerinum* is one of the Actinobacteria that are beneficial to human health as probiotics [59]. 

The addition of tannin-rich diets or tannin extracts alters rumen fermentation. It decreases CH_4_ production by directly inhibiting methanogens and indirectly decreasing H_2_ production due to reduced fiber digestion and protozoal population in the rumen [60,61]. It has been shown that tannin extracts (CT + HT) can reduce methanogens and protozoa populations in the rumen [62] so that CH_4_ emissions are decreased by 30–57% [13]. The alteration of methanogens by tannins is possibly due to interactions between a tannin molecule and the specific microbial cell walls to which it binds based on hydrophobic and hydrogen bonding in a pH–dependent manner [63,64]. Currently, many types of individual tannins have been purified and tested for their anti-microbial activity [65,66]. However, in the present study the *Methanobrevibacter* sp. (archaea) population had no interactions (*p* = 0.99) with the source of tannins or with the dosage of tannins (Table 3). Therefore, clear structure–anti-methanogenic activity patterns are required to explore the specific effects of tannin inclusions on fecal microbial activity and GHG emissions. 

At the genus level, 356 bacteria were detected, and the taxonomic analysis discovered the occurrence of 20 main genus bacteria in the fecal manure (Figure 1a,b). Alkaliflexus, Mycoplana, Planctomyces, Protiniphilum, and Clostridium genera accounted for 20.4%, 6.5%, 6.4%, 4.3%, and 3.5% of the total reads in control groups, respectively. Relative abundances of Mycoplana (*p* < 0.02), Proteiniphilum (*p* < 0.01), Clostridium (*p* < 0.01), Nonomuraea (*p* < 0.01), and Pusillimonas (*p* < 0.01) decreased for tannin groups versus control groups, while the relative proportion (%) of Shigella (*p* < 0.01), Corynebacterium (*p* < 0.01), Acinetobacter (*p* < 0.01), and Glycomyces (*p* < 0.01) significantly increased, which is a similar trend seen in current tannins studies with bacterial phyla and species levels. These results indicate that the plant tannins elicited a collective effect on the bacterial population and correspondingly suggest a reduction and/or enhance in the population of fecal microorganisms, as demonstrated by CH_4_ and N_2_O productions in these experiments.

### 3.3. Rates of Emissions and Cumulative Production of Methane (CH_4_) and Nitrous Oxide (N_2_O) 

There is considerable interest in understanding the effects of livestock manure management on direct and indirect sources of GHG emissions, as manures contain significant amounts of N, C, and water: these three necessary elements govern processes essential to emissions of CH_4_ and N_2_O [67,68]. The most of the N_2_O from manure is produced in manure-amended soils through microbial nitrification (converting NH_4_^+^ or organic N to NO_3_^−^ and NO_2_^−^) under aerobic circumstances and partial denitrification (converting NO_3_^−^ and NO_2_^−^ to N_2_O and eventually N_2_ through multiple redox reaction steps) under anaerobic settings [67,68,69,70,71]. Tannins are known to slow down the decomposition of organic materials [21] and may reduce nutrient losses and GHG emissions [61,72]. The effect of adding 4 and 8% of both CT and HT on the CH_4_ and N_2_O emissions of dairy cattle manure during in vitro incubation was measured, and the results are summarized in Figure 2, Figure 3, Figure 4 and Figure 5. With the addition of tannins, CH_4_ production markedly varied between two types of tannins. The addition of quebracho CT significantly reduced (*p* < 0.001) the CH_4_ flux rates (Figure 2) compared to the no-tannin controls (215.9 mg/m^2^/h), with peaks as high as 75.6 and 89.6 mg/m^2^/h for 4 and 8% CT, respectively. Furthermore, the CT treatments significantly reduced (*p* < 0.01) cumulative CH_4_ emission by 68.2 and 57.3% at 4 and 8% CT, respectively, with tannins, dose, and tannins x dose interactions (*p* < 0.01) (Figure 3). Results from previous and current research suggest that quebracho CT is effective in reducing selected microbial diversity (Table 2 and Table 3 and Figure 1a,b) and CH_4_ production [73,74]. These observations are consistent with other studies which showed that different levels of quebracho CT extracts in an in vitro addition [tannin-containing diets including sainfoin (*Onobrychis viciifolia*), birdsfoot trefoil (*Lotus corniculatus*), and sericea lespedeza (*Lespedeza cuneata*) CT inhibited the growth of rumen bacteria [61,72,75,76] and CH_4_ emissions in meat goats [13,18,19]. It has been reported that tannin reduces CH_4_ emissions by directly inhibiting methanogens [19,77] and ruminal microbiota that produces H_2_ and are associated with methanogens and protozoa [62,78], or indirectly by reducing forage diet degradation in animals [16,79]. Vasta et al. [79] and Min and Solaiman [14] hypothesized that plant tannins could directly inhibit CH_4_ production through decreased methanogenesis and reduced activities of selected rumen microbes (such as cellulolytic bacteria and protozoa) that modify the conversion of substrate to H_2_ and acetates. Consistent with Hao and Benke [40], the anti-methanogenic effect of tannins depends on the tannin’s concentration and is positively correlated to the number of hydroxyl groups in their structure. This study determined that tannins reduced only numerical numbers and need further investigation on the anti-methanogenic and anti-microbial activity of plant tannins [52,80,81,82].

The addition of HT showed a significant increased (*p* < 0.001) in manure-derived cumulative CH_4_ production of 30.5% at the low level of HT (4% HT), but the 8% HT treatment did not differ from the control (Figure 3). This agrees with data from Min et al. [55] who reported some rumen bacterial strains (*Clostridium proteoclasticum* B316^T^ and *Ruminococcus albus*) showed transient increases in their growth rate at low concentrations (50–100 µg/mL) of CT extracted from birdsfoot trefoil, but not at high (>200 µg/mL) concentrations of CT. Even though Bacteroidetes and Proteobacteria populations in dairy manure have decreased when incubated with CT and HT, the current study showed that the dairy cattle manure exhibited a more significant proportion of Actinobacteria when compared to the control. In addition, this enhancement of CH_4_ production at low levels of HT (<4% HT) may not be caused only by microbial population changes but also by structural changes in the substrates of carbohydrate and protein through its interaction with HT [8], allowing easier access of energy and protein sources [82]. This indicates that under certain conditions (e.g., pH, binding capacity, tannins:protein ratio), interactions between tannins and protein can generate soluble complexes, and Mole and Waterman [81] demonstrated the occurrence of inhibitory, stimulatory, and null effects on the proteolysis of these complexes [81]. Although CT decreased CH4 production, chestnut HT had little effect on CH_4_ production even at 8% HT. These observations were consistent with other studies showing that the addition of chestnut HT had little effect on in vitro [82] and in vivo CH_4_ production in beef cattle [83]. Quebracho CT is mainly comprised of dimers to polymers of (epi) catechin and profisetinidins, but chestnut HT contains mainly castalin/vescalin and castalagin/vescalagin tannins which correspond to gallotannins and ellagitannins [63,64,82]. HT substantially differs in water solubility [63,64] and exhibits weak interactions with proteins [83,84]. These structural differences likely contribute to the effects of these tannins on gas production and microbial community changes in dairy cattle manure. Furthermore, the great abundance of Bacteroidetes and Proteobacteria in the control groups of this present study suggests that these shifts of the microbiome associated with CT fractions may play a role in affecting GHG emissions as well as energy efficiency [26,85].

Evaluating N_2_O emissions from manure is a common way to detect a contrary relationship between CH_4_ and N_2_O emissions [82,86]. This present study revealed that N_2_O emission fluxes were negative (consumptive), indicating N_2_O movement from the headspace into the manure when the manure moisture levels were highest (wet and anaerobic condition) and in an early stage of fermentation up to day 9. Gradually, N_2_O emissions stabilized after 10 d as the levels of manure moisture decreased (Figure 4). Compared to N_2_O emissions in Figure 5, the CH_4_ flux peaked during the early fermentation stage on day 0 during the highest moisture content (anaerobic condition) and was significantly depressed by day 2. It has been confirmed that the occurrence of water in the soil is negatively related to O_2_ content. In the present study, the effect of incubation time (days) on moisture content was not tested. It has been reported that moisture contents influenced N2O emissions. The most significant emissions occurred at 80 to 100% water-filled porosity space (WFPS; [87]) or when the moisture content was between 40 and 60% with 10% oxygen content [88]. It has been reported that N_2_O released by soils can be produced either by denitrification in anoxic circumstances or by nitrification in the presence of O_2_ [89,90]. When the soil is near complete water saturation, anaerobic conditions prevail and N_2_O emission drops, but N_2_ emission rises sharply as a result of denitrification [88]. The aerobic microenvironment of the dried aerated crust preferred the growth of bacterial communities capable of nitrification and denitrification in aerobic/anaerobic sequestered microsites, which mutually are the reactions responsible for the formation of N_2_, N_2_O, and NO_3_^-^ [86]. Hwang and Hanaki [88] reported that as the bulk space of the manure became aerobic, the total amount of N_2_ produced from denitrification decreased, but the proportion of N_2_O increased. Recently, Parker et al. [91] and Waldrip et al. [25] confirmed that the addition of simulated rainfall into dry beef cattle manure caused an immediate increase in N_2_O-N flux (N_2_O as N) from the laboratory chambers, with peaks as high as 7.0 mg N/m^2^/h. However, significant manure characteristics, such as NH_4_^+^ and NO_3_^−^ concentrations, manure DM content, redox status, and temperature, influenced N_2_O losses. Waldrip et al. [25] reported that most feedyard N_2_O emissions were derived from denitrification in the top 5 cm of the feedlot manure pack. The flux of N_2_O in the present study obtained from the dairy cattle manure interface results from dynamic production and consumption processes in the manure nitrification and denitrification processes. However, the factors regulating N_2_O production associated with plant tannins are not yet well understood and merit further study.

N_2_O fluxes (Figure 4) varied significantly between two types of tannin sources. Intermediate N_2_O reduction rates were detected for HT between −0.2 and −0.4 mg/m^2^/h from d 0 to d 4. The addition of chestnut HT at 4% negatively impacted (*p* < 0.001) cumulative N_2_O sinks (−79.05 mg/m^2^/h) when compared to the control (−50 mg/m^2^/h), 8% HT addition (−49.78 mg/m^2^/h) and/or 8% CT addition (−39.62 mg/m^2^/h) over the 14-d period (Figure 5), indicating that HT may affect more N_2_O emission in dairy cattle manure compared to CT [92]. The HT treatments failed to affect CH_4_ reduction. However, both CT and HT reduced (*p* < 0.001) cumulative and flux rates of N_2_O emissions. The literature has reported that the maximum net negative N_2_O fluxes vary widely, from −0.14 to −0.48 mg/m^2^/h [93]. This indicates that dairy cattle manures are usually considered sources of net N2O emissions, but they can also act as sinks, similar to soils [93]. The sink strength depends on the potential for N_2_O reduction to N_2_, the ease of N_2_O diffusion within the soil profiles, and its dissolution in soil water [93]. In the presence of CT and HT, CH_4_ and N_2_O production and the interrelationships of those GHG productions might be connected to specific changes in predominant microflora associated with supplementing tannins in the fermentation chamber (Figure 6; [37,94]). Therefore, both CT and HT could regulate N_2_O emissions associated with nitrification and denitrification by inhibiting decomposition and microbial activity [21,95].

## 4. Conclusions

In this research, the mitigating effect of tannin-based treatment in dairy cattle manure on CH_4_ and N_2_O emissions was investigated. The CT applied directly to the dairy cattle manure reduced CH_4_ emissions compared with the control group, while HT failed to affect CH_4_ reduction. However, both CT and HT strongly negatively affected N_2_O emissions. These results show that tannins are a promising method for reducing GHG emissions from dairy cattle manure, particularly under laboratory conditions. Furthermore, in contrast to chestnut HT, quebracho CT more effectively reduced CH_4_ emissions, reflecting a potential decline in Bacteroidetes and Proteobacterial phyla populations. These results suggested that the 4% CT inclusion is a promising technique for reducing CH_4_ emissions in dairy cattle manure composting. Field-scale studies are currently needed to determine if using quebracho CT and chestnut HT in dairy cattle manure can significantly reduce CH_4_ and N_2_O production and methanogenesis under commercial conditions. Further studies are also necessary to investigate the long-term effects of lowering N_2_O emissions and clarify the transformation process of nitrogen in the soil when complexed to tannin-based dairy cattle manure treatment.

## Figures and Tables

**Figure 1 animals-12-02876-f001:**
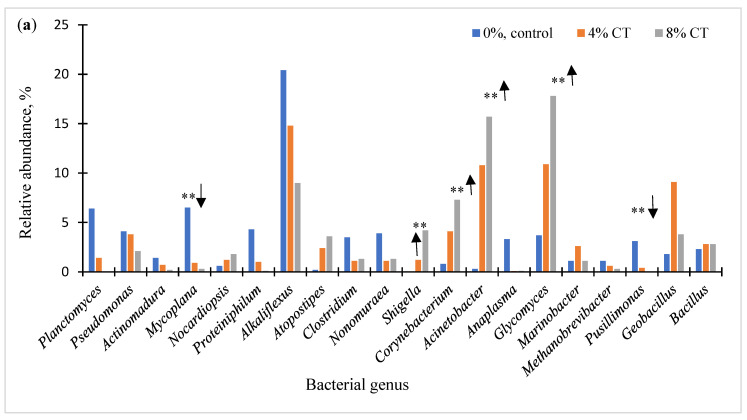
Genus-level classification of the bacterial community composition (n = 3) in condensed (CT; (**a**)) and hydrolysable tannins (HT; (**b**); relative abundance >1.0%). Asterisks (**) represent significance levels of *p* < 0.01 with increased (↑) or decreased (↓) bacterial abundance.

**Figure 2 animals-12-02876-f002:**
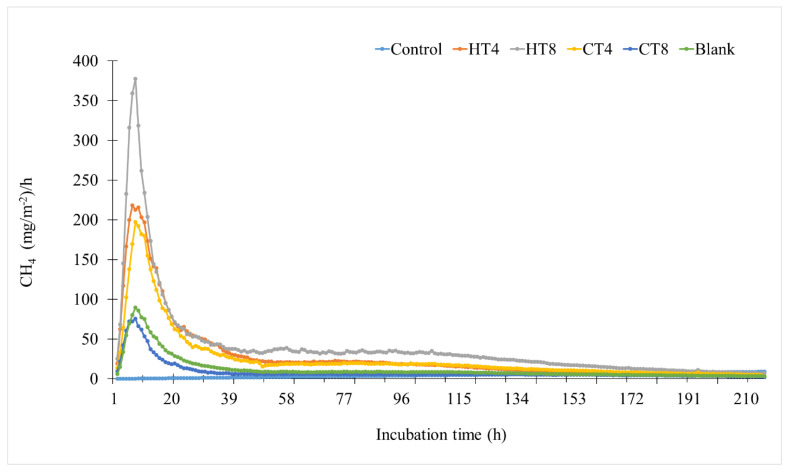
Hourly rates of methane emissions during in vitro incubation (n = 3) with dairy cattle manure composting. CT = quebracho condensed tannins; HT = chestnut hydrolysable tannins.

**Figure 3 animals-12-02876-f003:**
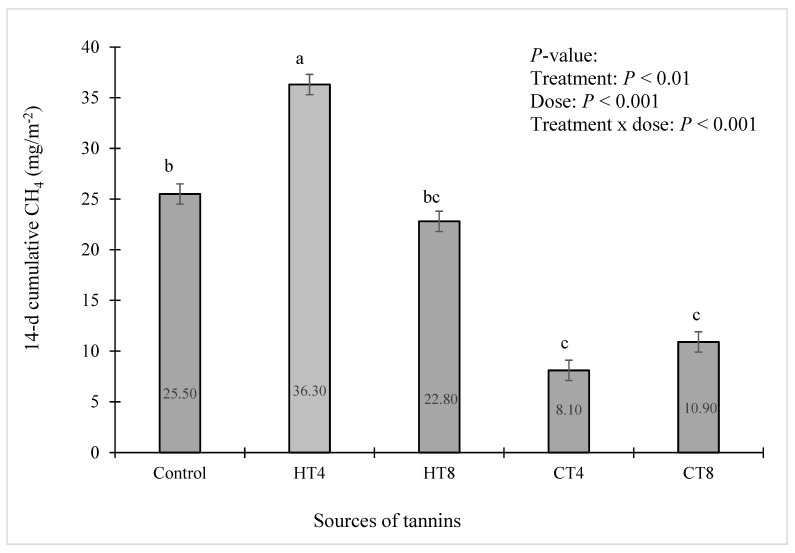
The effect of sources of tannins on cumulative CH_4_ emissions from in vitro incubation with dairy cattle manure for 14 d. Data show means (*n* = 3), numbers in a row followed by different lowercase letters (^a–c^) indicate treatment effect (*p* < 0.05–0.001). CT = quebracho condensed tannins (4% and 8%); HT = chestnut hydrolysable tannins (4% and 8%). Error bars represent the standard error of the means.

**Figure 4 animals-12-02876-f004:**
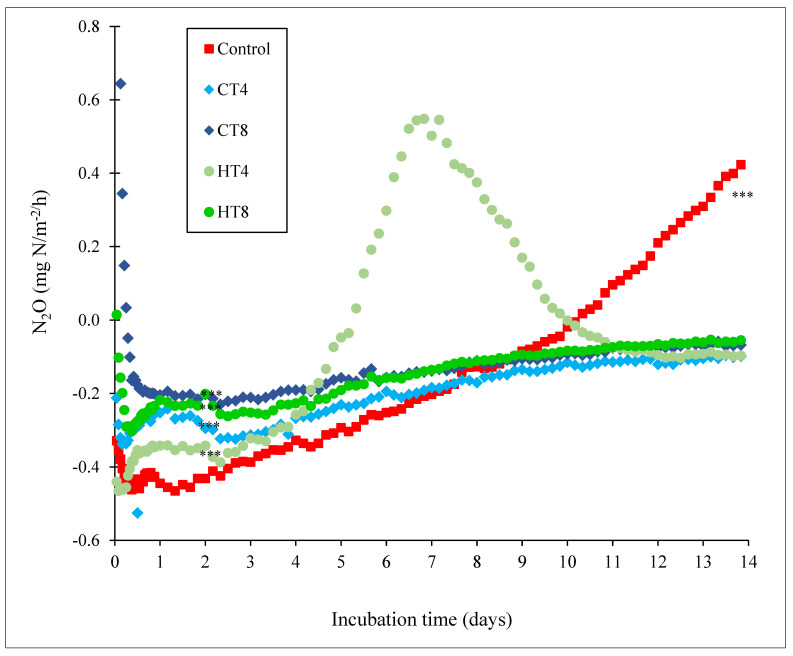
Hourly rates of nitrous oxide (N_2_O) emission during in vitro dairy cattle manure incubation (n = 3) experiment for 14 d composting period. CT = quebracho condensed tannins; HT = chestnut hydrolysable tannins. All treatments were a sink (negative emissions) except for the control. Asterisks (***) represent significance levels of *p* < 0.001.

**Figure 5 animals-12-02876-f005:**
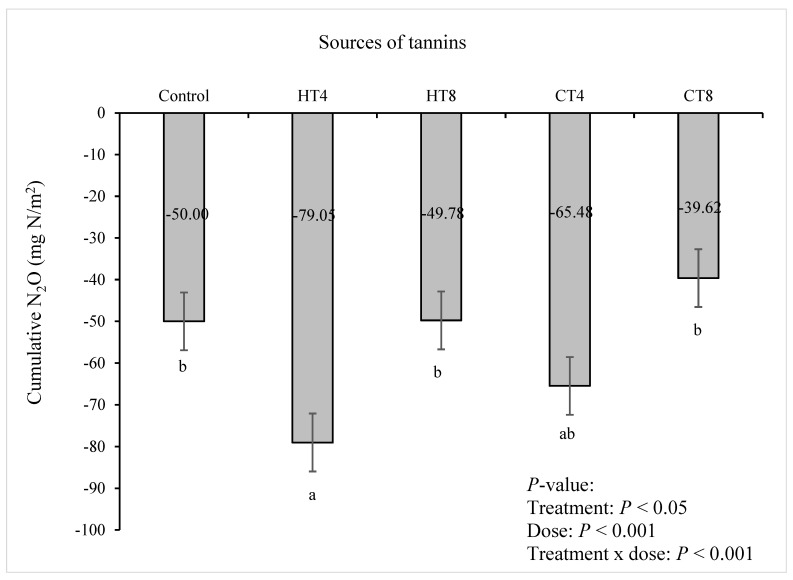
The effect of sources of tannins on cumulative nitrous oxide (N_2_O) emissions from in vitro incubation with dairy cattle manure for 14 d. Data show means (*n* = 3), numbers in a row followed by different lowercase letters (^a,b^) indicate treatment effect (*p* < 0.05). *R*^2^ = 1.44. CT = quebracho condensed tannins (4% and 8%); HT = chestnut hydrolysable tannins (4% and 8%). Error bars represent the standard error of the means.

**Figure 6 animals-12-02876-f006:**
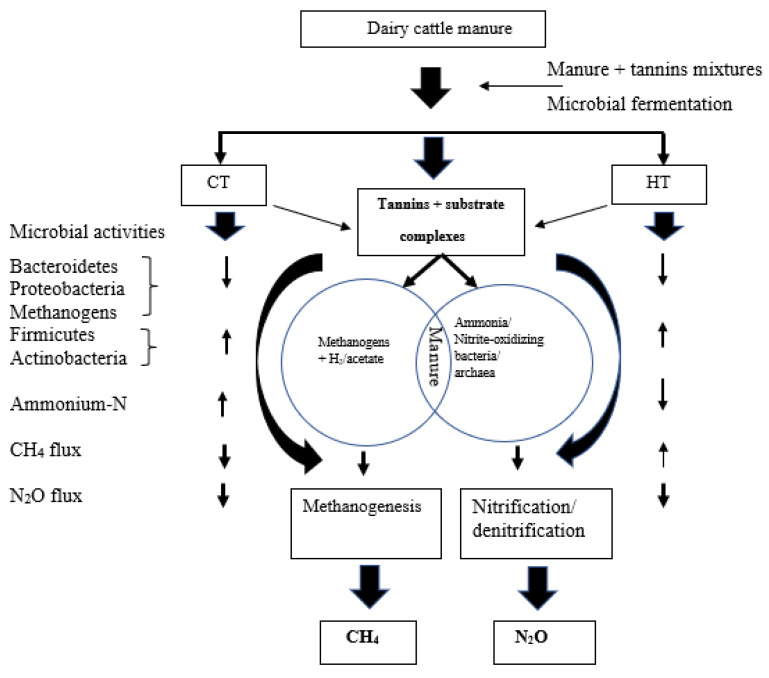
Proposed effect of condensed (CT) and hydrolyzable (HT) tannins on fecal bacteria and substrates and their interactions. Sources: current study; Barnard et al. [96]; Patra and Saxena [60]; Cáceres et al. [96]; and Min et al. [61]. Arrow bars represent the increase or decrease in microbial activities, ammonia-N, methane (CH_4_), and nitrous oxide (N_2_O) flux associated with CT and HT.

**Table 1 animals-12-02876-t001:** Chemical composition of experimental dairy cattle manure at day 0 and day 14.

		Quebracho CT		Chestnut HT	
Item	Control	4% CT	8% CT	4% HT	8% HT
Chemical composition at day 0					
Moisture, %	62.7	61.9	61.9	62.3	59.8
Total solids, as-is basis	37.3	38.1	38.1	37.7	40.2
	------------------------- % DM ------------------------------				
Ash, % DM	7.8	6.9	7.1	7.5	7.5
Organic matter, % DM	29.5	31.2	31.0	30.2	32.8
Total N, % DM	9.4	8.5	8.4	9.1	8.5
Organic-N, % DM	9.2	8.5	8.3	9.1	8.5
C/N ratio	18.3	21.3	21.4	19.2	22.2
Ammonium–N, mg/kg	0.02	0.03	0.06	0.06	0.03
Nitrate-N, mg/kg	10.9	10.9	10.0	10.8	10.9
Nitrite–N, mg/kg	1.0	1.0	1.0	0.9	0.9
Nitrate-N + Nitrite-N, mg/kg	11.9	11.9	11.0	11.7	10.9
pH	8.8	8.5	8.5	7.8	7.1
Chemical composition at day 14	------------------------- % DM ----------------------------				
Moisture, %	29.5	23.6	26.4	30.7	31.8
Total N, % DM	1.8	1.7	1.7	1.9	1.6
Organic-N, % DM	1.8	1.7	1.6	1.8	1.6
Ammonium-N, mg/kg	0.19	0.42	0.72	0.25	0.25
Nitrate-N, mg/kg	0.63	0.73	0.65	0.71	0.71
Nitrite-N, mg/kg	1.0	1.0	1.0	0.9	0.9
Nitrate-N + Nitrite-N, mg/kg	1.63	1.73	1.65	1.61	1.61
N loss or retention	------------------------- % -------------------------------				
Total N	−80.9	−80.0	−79.8	−79.1	−81.2
Organic-N	−80.4	−80.0	−80.7	−80.2	−81.1
Ammonium-N	89.4	92.9	91.7	76.0	88.0
Nitrate-N	−94.2	−93.3	−95.7	−93.4	−98.7
Nitrite-N	0	0	0	0	0
Nitrate-N + Nitrite-N	−86.3	−85.5	−85.0	−86.2	−85.2

CT = condensed tannins; HT = hydrolysable tannins; DM = dry matter; N = nitrogen. Total solids, ash, organic matter, total N, NH_4_^+^-N, NO_3_-N, and pH on d 0 were similar among treatment groups, but C/N ratio was generally greater (*p* < 0.05) for CT and HT groups than for the control group (Table 1). Only trace amounts of NH_4_^+^-N were detected in dairy cattle manure. After manure was incubated, with or without tannins, at 14 d, total N (%), organic-N (%), NH_4_-N, and NO_3_^-^/NO_2_^-^ (mg/kg) were similar among treatments (Table 1). In the present study, the total N and nitrate+ nitrite losses tended to be lower (*p* = 0.11) during 14 d of composting (Table 1). The diet in dairy ration contains a minimum of 27% neutral detergent fiber (NDF) or 19% acid detergent fiber (ADF) (DM basis), with 75% of the ration NDF-derived from forage/roughage (generally winter wheat silage and mixed grass hay).

**Table 2 animals-12-02876-t002:** Mean relative abundance values (%) of the most predominant bacterial phyla (> 1.0%) as a function of the addition of tannins with different dose levels of tannins in dairy cattle manure.

Item	Bacterial Phylum, %						*p*-Value		
	Control		CT, %	HT, %					
	0.0	4.0	8.0	4.0	8.0	SEM	Tannins	Dose	Interaction
Bacteroidetes	28.8 ^a^	18.9 ^ab^	10.5 ^b^	30.4 ^a^	9.4 ^b^	4.78	0.04	0.03	0.13
Firmicutes	14.1	24.9	27.0	17.6	28.4	5.54	0.12	0.36	0.60
Proteobacteria	32.8	29.0	28.4	21.1	23.6	4.30	0.09	0.86	0.93
Actinobacteria	13.7 ^b^	23.6 ^ab^	32.6 ^a^	22.2 ^ab^	33.6 ^a^	4.09	0.01	0.05	0.35
Planctomycetes	7.0	1.4	0.1	6.5	1.0	3.45	0.23	0.43	0.72
Euryarchaeota	1.1	0.6	0.4	0.5	1.5	0.76	0.67	0.67	0.71
Chloroflexi	1.3	0.8	0.4	1.0	1.0	0.57	0.50	0.80	0.94

^a,b^ Means within row treatment with a different superscript differ at *p* < 0.05. CT = condensed tannins; HT = hydrolysable tannins.

**Table 3 animals-12-02876-t003:** Mean relative abundance values (%) of the most predominant bacterial species (>1.0%) as a function of the addition of tannins with different dose levels of tannins in dairy cattle manure.

Item Phylum/Species	Bacterial Species, %						*p*-Value		
	Control		CT, %	HT, %					
	0.0	4.0	8.0	4.0	8.0	SEM	Tannins	Dose	Interaction
Firmicutes, %									
*Bacillus* spp.	4.3	5.1	4.9	8.2	1.8	3.19	0.96	0.41	0.54
*Clostridium* spp.	3.4 ^a^	1.0 ^b^	1.2 ^b^	1.1 ^b^	2.6 ^a^	0.47	0.01	0.17	0.25
Bacteriodetes, %									
*Proteiniphilum* spp.	4.3 ^a^	1.1 ^b^	0.1 ^b^	6.4 ^a^	1.5 ^b^	0.86	0.001	0.01	0.03
*Alkaliflexus* sp.	20.4 ^a^	14.8 ^a^	9.0 ^b^	20.6 ^a^	4.2 ^b^	4.15	0.11	0.05	0.17
Proteobacteria, %									
*Pusillimonas* *noertemannii*	2.0 ^a^	0.09 ^b^	0.01 ^b^	0.1 ^b^	0.04 ^b^	0.54	0.01	0.92	0.99
*Pseudomonas* *tuomuerense*	2.1	0.3	0.2	1.0	1.9	0.76	0.08	0.66	0.78
*Shigella sonnei*	0.05 ^b^	1.2 ^ab^	4.2 ^a^	0.1 ^ab^	2.4 ^a^	0.86	0.03	0.02	0.23
*Idiomarina indica*	2.5 ^a^	0.7 ^b^	0.01 ^b^	0.06 ^b^	0.02 ^b^	0.15	0.001	0.08	0.09
Actinobacteria, %									
*Nonomuraea* sp.	3.8 ^a^	0.3 ^b^	0.05 ^b^	5.4 ^a^	0.7 ^b^	0.66	0.001	0.01	0.01
*Glycomyces* sp.	3.5	5.4	5.9	4.4	3.6	1.88	0.40	0.74	0.94
*Bifidobacterium* *choerinum*	0.02 ^b^	0.5 ^ab^	1.0 ^a^	0.04 ^b^	0.6 ^ab^	0.32	0.05	0.19	0.63
Tenericutes, %									
*Mycoplana* spp.	6.5 ^a^	0.9 ^b^	0.3 ^b^	3.4 ^a^	0.8 ^b^	2.23	0.05	0.56	0.83
*Corynebacterium* sp.	0.5 ^c^	2.0 ^b^	5.0 ^a^	0.5 ^c^	2.9 ^b^	1.19	0.05	0.12	0.53
Planctomycetes, %									
*Planctomyces* spp.	6.5	1.4	0.03	6.4	0.9	3.23	0.25	0.40	0.68
Euryarchaeota (Archaea), %									
*Methanobrevibacter* sp.	1.1	0.4	0.4	0.5	0.3	0.57	0.42	0.88	0.99

^a–c^ Means within row treatment with a different superscript differ at *p* < 0.05. CT = condensed tannins; HT = hydrolysable tannins.

## Data Availability

The datasets generated and/or analyzed during the current study are available from the corresponding author (B.R.M.) upon reasonable request.

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
