# Peer review of "Condensed and Hydrolyzable Tannins for Reducing Methane and Nitrous Oxide Emissions in Dairy Manure—A Laboratory Incubation Study†"

_animals, 2022, doi:10.3390/ani12202876_

Round 1

Reviewer 1 Report

Line 53, replace monogastric by non-ruminants

Line 70, eliminate “This is of interest” , leave to reader to decide.

Lines 455 to 464, this paragraph looks as conclusion than a discussion. I recommend to reduce it or eliminate. 

Author Response

Reviewer 1

Line 53, replace monogastric by non-ruminants: Changed to non-ruminants.

Line 70, eliminate “This is of interest” , leave to reader to decide. Eliminated “This is of interest”

Lines 455 to 464, this paragraph looks as conclusion than a discussion. I recommend to reduce it or eliminate: Deleted a sentence.

Reviewer 2 Report

The text in the sentence does not fit with the results (row 34, 35 in Abstract; 438, 439 in Results and Discussion; 469 in Conclusion)  

Materials and Methods: add composition of diets, is needs to specify dairy cattle, add determination of organic matter content

Results and Discussion: 220- 19 phyla (this is not apparent from Table 2); presented results from Table 2- in Table 2 is 0.05, but in the text is ˂0.05 (correct it);
276 and 277- correct font size; 251 and 296- the numbers cannot be verified;
300 and 301- text is not in accordance with the Figure 1a and 1b; explain under Figure 1. the meaning of (a) and (b); Figure 2- QT4, QT8 change to CT4, CT8;
366- Firmicutes were not statistically significant; 390- 62 % it's not an exact number; 396- ,,Moisture contents influenced the N2O emissions,, in the manuscript is effect of tannins addition on moisture content, but the effect of incubation time (days) on moisture content is not mentioned; 435 and 436 the numbers put in accordance with the Figure 5; 36 and 473- Proteobacterial phyla population were not statistically significant affected

In the text references 75, 76, 85, 89 are missing.

Author Response

Reviewer 2: Comments and Suggestions for Authors

The text in the sentence does not fit with the results (row 34, 35 in Abstract; 438, 439 in Results and Discussion; 469 in Conclusion): Corrected the sentence.

Materials and Methods: add composition of diets, is needs to specify dairy cattle, add determination of organic matter content.  Added a sentence under the table 1 “The diet in dairy ration contains a minimum of 27% neutral detergent fiber (NDF) or 19% acid detergent fiber (ADF) (DM basis), with 75% of the ration NDF derived from forage/roughage (generally winter wheat silage and mixed grass hay).

Results and Discussion:

220- 19 phyla (this is not apparent from Table 2); seven major bacterial phyla (relative abundance > 1.0%) as dominant, regardless of the treatment group.

presented results from Table 2- in Table 2 is 0.05, but in the text is Ë‚0.05 (correct it);Changed to P<0.04

276 and 277- correct font size; 251 and 296- the numbers cannot be verified; Changed to correct number.

300 and 301- text is not in accordance with the Figure 1a and 1b; explain under Figure 1. the meaning of (a) and (b); These results indicate that the condensed tannins elicited a collective effect on the bacterial population and correspondingly suggest a reduction in the population of methanogenic microorganisms, as demonstrated by reduced CH4 production in these experiments.

 Figure 2- QT4, QT8 change to CT4, CT8;Corrected in Figure 2

366- Firmicutes were not statistically significant; Deleted Firmicutes.

 390- 62 % it's not an exact number; Deleted 62%

396- ,,Moisture contents influenced the N2O emissions,, in the manuscript is effect of tannins addition on moisture content, but the effect of incubation time (days) on moisture content is not mentioned; In the present study, the effect of incubation time (days) on moisture content is not tested. It has been reported that moisture contents influenced the N2O emissions

 435 and 436 the numbers put in accordance with the Figure 5; Corrected the numbes.

36 and 473- Proteobacterial phyla population were not statistically significant affected

In the text references 75, 76, 85, 89 are missing. Fixed the all the references

Reviewer 3 Report

Interesting manuscript, not too innovative topic. However, this manuscript also shows the importance of reproducibility of similar research, because regardless of the theoretical basis of GHG reduction with tannins, in experiments and in practice the results are unexpected and inconsistent.

Line 96: The sentence needs to be reformulated, it is not exactly that the sample was taken from all 2000 cows, as can be read from this sentence.

Line 137: Explain why you took the sample this way, rather than simply mixing the whole mass.

Line 184-202: You do not need to state "Table 1" in every sentence, it is enough in one place.

In Table 1, you do not have any statistically significant differences between the treatments?

Line 240: „actinomycetes“ font !!!

Line 392: „decrease after 10-d“, as can be seen on Figure 4, with both 8-treatments, there is still an increase.

Line 469: „both CT and HT strongly inhibitory affected N2O emmision.“ How did you conclude that? From Figure 5, you can't really see that, that is, the representation is different from your claim.

Why is the HT4 treatment so different to you? Nowhere in the text did you refer to that issue, did any error occur in the gas measurements?

References: You have too much line spacing!

Author Response

Reviewer 3

Line 96: The sentence needs to be reformulated, it is not exactly that the sample was taken from all 2000 cows, as can be read from this sentence. Added “Twenty fresh dairy cattle manure was obtained by randomly collecting”

Line 137: Explain why you took the sample this way, rather than simply mixing the whole mass. This is because of same sampling procedures with other our feedlot field study in Panhandle area in Texas.

Line 184-202: You do not need to state "Table 1" in every sentence, it is enough in one place. Deleted Table 1 in other text.

In Table 1, you do not have any statistically significant differences between the treatments?

In recent years, many analyses have been carried out to investigate the chemical components of food and feed data. However, studies rarely consider the compositional pitfalls of such analyses. This is problematic as it may lead to arbitrary results when non-compositional statistical analysis is applied to compositional datasets. 

Line 240: „actinomycetes“ font !!! Corrected.

Line 392: „decrease after 10-d“, as can be seen on Figure 4, with both 8-treatments, there is still an increase. Changed to “stabilized after 10-d”

Line 469: „both CT and HT strongly inhibitory affected N2O emmision.“ How did you conclude that? From Figure 5, you can't really see that, that is, the representation is different from your claim. Changed to “negatively affected N2O emissions”

Why is the HT4 treatment so different to you? Nowhere in the text did you refer to that issue, did any error occur in the gas measurements? Low levels of Hydrolysable tannins (HT$) possibly supply energy to the fecal microbiomes somehow to transit increases of gas production, but not in higher levels (HT8), Min et al., 2005b [55].

References: You have too much line spacing! Checked the references
